# Microbiome Therapeutics for Food Allergy

**DOI:** 10.3390/nu14235155

**Published:** 2022-12-03

**Authors:** Diana A. Chernikova, Matthew Y. Zhao, Jonathan P. Jacobs

**Affiliations:** 1Department of Pediatrics, Division of Immunology, Allergy, and Rheumatology, David Geffen School of Medicine at UCLA, Los Angeles, CA 90073, USA; 2The Vatche and Tamar Manoukian Division of Digestive Diseases, David Geffen School of Medicine at UCLA, Los Angeles, CA 90095, USA; 3Division of Gastroenterology, Hepatology and Parenteral Nutrition, Veterans Affairs Greater Los Angeles Healthcare System, Los Angeles, CA 90073, USA

**Keywords:** microbiome, food allergy, live microbial therapeutics, bacteriotherapy, probiotics, metabolites

## Abstract

The prevalence of food allergies continues to rise, and with limited existing therapeutic options there is a growing need for new and innovative treatments. Food allergies are, in a large part, related to environmental influences on immune tolerance in early life, and represent a significant therapeutic challenge. An expanding body of evidence on molecular mechanisms in murine models and microbiome associations in humans have highlighted the critical role of gut dysbiosis in the pathogenesis of food allergies. As such, the gut microbiome is a rational target for novel strategies aimed at preventing and treating food allergies, and new methods of modifying the gastrointestinal microbiome to combat immune dysregulation represent promising avenues for translation to future clinical practice. In this review, we discuss the intersection between the gut microbiome and the development of food allergies, with particular focus on microbiome therapeutic strategies. These emerging microbiome approaches to food allergies are subject to continued investigation and include dietary interventions, pre- and probiotics, microbiota metabolism-based interventions, and targeted live biotherapeutics. This exciting frontier may reveal disease-modifying food allergy treatments, and deserves careful study through ongoing clinical trials.

## 1. Introduction

### 1.1. Food Allergy

Food allergies have been rising in prevalence in recent decades and are also the most common cause of anaphylaxis in children [1,2]. Anaphylaxis is a serious allergic reaction that can be life-threatening and involves various organ systems, including the respiratory tract, gastrointestinal tract, and skin; it is primarily treated with epinephrine administration [3]. Food allergies impose significant burdens on patients and families due to the need for specialized diets and constant monitoring for allergens in food, increased healthcare usage, and anxiety related to developing an anaphylactic reaction [4,5,6]. The main treatments for food allergies include allergen avoidance, treatment of allergic reactions with epinephrine and other medications, and oral immunotherapy [7,8]. Oral immunotherapy involves oral administration of a food allergen, either in fixed doses or in gradual doses until a maintenance dose is reached [9]. The goal of oral immunotherapy is to desensitize a patient against a food allergen; that is, to increase the threshold dose needed for a patient to develop an allergic reaction to food. There is only one FDA-approved treatment for food allergies, and it is the peanut oral immunotherapy called Palforzia [10]. However, oral immunotherapy is not a cure for food allergies, and discontinuation of immunotherapy usually results in a loss of tolerance to the allergen. In addition to oral immunotherapy, other immunotherapies for food allergies exist, including sublingual and epicutaneous immunotherapy. Oral immunotherapy is the most effective of these, but comes with high rates of adverse events, including allergic reactions and the development of eosinophilic esophagitis [8]. Adjunct therapies to oral immunotherapy to help decrease adverse reactions include the use of monoclonal antibodies, such as omalizumab (anti-IgE) and dupilumab (anti-IL-4Rα) [11,12,13]. Given the few treatment options for food allergies, there is a substantial interest in and need for the new strategies to prevent and treat food allergies.

Food allergies are thought to arise due to a combination of genetic and environmental factors [14,15]. Genetic association studies have identified some risk alleles for food allergies, particularly in the genes *MALT1*, *FLG*, and *HLA*; these findings implicate genes involved in immune and barrier function in the development of food allergies [16]. Environmental exposures have also been associated with allergy development [17,18,19,20]. For example, it has been observed that different dietary exposures can lead to different rates of food allergy acquisition, as seen in discordant peanut allergy rates among genetically similar populations in Israel and the UK who, notably, had different peanut consumption rates [21,22]. Furthermore, exposure to the farm environment and pets is associated with a decreased risk of allergy development, while exposure to antibiotics and the Western diet increases future allergy risk [18,23,24,25,26].

### 1.2. Gut Microbiota and Food Allergy

Recent evidence implicates the composition of microbes in the gut (the “microbiome”) as a critical mediator of the relationship between environmental factors and the development of food allergies and other allergic diseases, such as atopic dermatitis and asthma [27,28]. Some of the first studies to suggest the importance of the gut microbiota in modulating food allergies were performed using germ-free mice. These mouse studies demonstrated that oral tolerance to food allergens was not achievable in germ-free mice, and that intestinal microbiota reconstitution was only successful in inducing oral tolerance when performed in neonatal mice [29]. Thus, not only are gut microbiota necessary for oral tolerance to food allergens, but their effects on the immune system were most strongly felt early in life. A few more recent studies have demonstrated that gut microbiota can also transmit susceptibility to food allergies when transferred from patients with food allergies to germ-free mice [30,31]. In observational human cohort studies, differences in gut microbiota composition have been found between subjects with food allergies compared to those without, suggesting that different microbiota may exhibit dissimilar effects on food allergen tolerance [32,33,34].

The relationship between the gut microbiome and food allergy development is believed to be mediated through microbial immune modulatory effects on food allergen tolerance [35]. One of the mechanisms by which the gut microbiome promotes tolerance is through induction of regulatory T-cells, which can be achieved through microbial production of short chain fatty acids, such as butyrate [17]. The intestinal microbiota may also prevent allergic sensitization to foods via induction of IL-22 production by immune cells, leading to decreased intestinal epithelial permeability and reduced interaction of the immune system with the allergen [36]. Given the extensive role for microbes in modulating food allergies, there is interest in modulating gut microbiome composition and function to prevent or treat food allergies.

### 1.3. Early Life Influences on Gut Microbiota and Immune Development

There are a variety of factors that affect the composition of the gut microbiome, which is highly variable in the first three years of life, after which it more closely resembles the adult gut microbiome [37,38,39]. At birth, the mode of delivery (vaginal delivery vs. caesarean section) shapes the initial microbial acquisition of the infant [40]. At that time, the child may be exposed to antibiotics administered directly or transferred across the placenta or through breastmilk after maternal administration of peripartum antibiotics [41,42]. Shortly afterwards, the infant’s diet will affect bacterial colonization, as the infant can be fed with breastmilk and/or formula [43]. Gestational prematurity is also thought to affect the microbiome, whether due to prematurity of the gut epithelium or the unique microbial exposures, antibiotics, and parenteral nutrition in the neonatal intensive care environment [44]. Later in life, the modern Western diet is believed to promote gut microbial dysbiosis, which may be contributing to the growing incidence of food allergies and autoimmune disease [45,46]. An important timepoint in the development of the gut microbiome is the weaning period, which is usually around age 4–6 months in human infants. There are rapid changes in microbial composition as the infant’s diet switches from exclusively milk-based to a diet incorporating a variety of solid foods [47]. In mice, there is a significant increase in gut microbial diversity at this time, which is called the “weaning reaction”. A study in mice showed that the time period of the weaning reaction is associated with imprinting of the immune system (through induction of regulatory T-cells); perturbations to the gut microbiota (via antibiotic administration) during that time were associated with a higher susceptibility to pathological inflammation [48]. This key timepoint can also been seen as a “window of opportunity”, as early introduction of peanuts during that time in human infants has been associated with a decreased risk of peanut allergy [49]. Additionally, the composition of the microbiome at that time has been associated with food allergy trajectory later in life; specifically, the presence of taxa from the Firmicutes phylum (which includes Clostridia) at age 3–6 months was associated with the resolution of milk allergy later in life [50,51]. Furthermore, in a study by Henrick et al., failure of infants to be colonized with bifidobacteria during the first months of life is associated with immune activation, decreased levels of regulatory T-cells, and increased intestinal inflammation [52].

Given the importance of microbiome composition at this timepoint, various approaches have been used to optimize gut composition in early life, including dietary interventions and probiotic administration (Figure 1). There is also significant interest in developing effective microbiome therapeutics with reproducible results [53]. In this review, we will discuss microbiome-related approaches to prevent or treat food allergies.

## 2. Dietary Approaches

A large variety of dietary interventions to prevent the development of food allergies have been studied. These include exclusive breastmilk feeding, use of elemental formulas in lieu of cow’s-milk based formulas, modifying the maternal diet during pregnancy and breastfeeding, early introduction of allergens, and consumption of fiber and a higher diversity of foods in early life [14,15,22,54,55,56]. Some benefit has been found in almost all of these approaches, but no approach has been successful in fully preventing food allergy development [57,58]. Some of the strongest evidence for benefit in prevention in food allergies is early introduction of peanut and egg, though it has also been found that avoidance of cow’s milk (via exclusive breastmilk feeding or supplementation with an elemental formula) in first 3 days of life may help decrease sensitization to cow’s milk [59,60,61]. Exclusive breastfeeding, while still strongly encouraged for various health benefits, does not appear to prevent food allergies [55,57]. Additionally, while the maternal diet during pregnancy has been associated with the development of various allergic diseases, it has not been shown to affect food allergy outcomes [62].

Prebiotics and probiotics for the prevention and treatment of various diseases have been widely studied, but results have been mixed [63]. While there have been reports of beneficial outcomes with the use of prebiotics and/or probiotics in food allergies, it has been difficult to make conclusions about the effects of these products given the wide variety of prebiotic and probiotic strains that have been studied, and the heterogeneity in dosages and administration [64,65,66,67,68,69]. Further studies are required to determine which prebiotics and probiotics, and at which doses, are effective in the prevention or treatment of food allergies.

## 3. Metabolites

Metabolites of bacterial fermentation and from dietary sources may have profound effects on host immunity and food hypersensitivity [15,70,71]. Emerging research has identified critical roles for immunomodulatory microbial metabolites in the development of food allergies in preclinical models.

### 3.1. Short Chain Fatty Acids

Dietary fiber undergoes metabolism by *Clostridium*, *Lactobacillus*, and *Bifidobacterium* species into short chain fatty acids (SCFAs), primarily acetate, butyrate, and propionate [72,73,74]. These metabolites bind SCFA-specific G-protein-coupled receptors, resulting in cell signaling to regulate eubiosis as well as inflammatory responses. Propionate binds GPR41, which is abundantly expressed in colonic epithelium, and the consumption of propionate has been shown to promote antigen-presenting cell precursors in a GPR41-dependent manner [75,76]. Furthermore, GPR43 is another SCFA-binding G-protein coupled receptor expressed in immune cells and colonic epithelium [75,77,78]. Existing GPR43 knock-out models exhibit increased susceptibility to food allergies; moreover, high fiber diets promote GPR43-dependent mechanisms to enhance tolerogenic CD103^+^ dendritic cells which, in turn, promote the differentiation of regulatory T-cells [79].

The SCFAs also modulate gene transcription via epigenetic mechanisms, such as the inhibition of histone deacetylases [80,81]. Butyrate has been found to protect against food allergies by promoting immune tolerogenic mechanisms, such as enhanced IL-10 expression and increased regulatory T-cell and IgA production, as well as by supporting mucosal integrity via increased expression of the goblet-cell mucin gene MUC2 [82,83,84]. Butyrate also inhibits IL-5 and IL-13 to modulate type 2 innate lymphoid cell-driven allergic inflammation [85].

In human studies, low levels of SCFAs have been associated with allergic symptoms in early life, and metagenomic analyses have shown that the microbiota of children who later develop allergic sensitization have reduced potential for butyrate production [86,87,88,89,90]. Variations in SCFA levels in the setting of gut dysbiosis have been linked to cow’s milk allergy, and enrichment with butyrate-producing microbiota promotes cow’s milk allergy resolution [50,91,92]. Oral supplementation with SCFAs did not protect against food allergies in a murine model (Il4raF709), although this supplementation was a combination of acetate, propionate, and butyrate in equal concentrations [30]. One clinical trial aims to investigate the use of adjuvant butyrate supplementation in oral peanut immunotherapy (ACTRN12617000914369). In this study, children aged 10–16 years with peanut allergy are randomized to receive peanut immunotherapy with or without a butyrylated high-amylase maize starch dietary supplement. Investigators will subsequently assess clinical tolerance to double-blind placebo-controlled food challenge at 13.5 months post-randomization, as well as sustained tolerance at 25.5 months post-randomization.

### 3.2. Secondary Bile Acids

Gut microbiota metabolize primary bile acids into various secondary bile acids, which, in turn, feedback to regulate the gut microbiome as well as subsequent cellular signaling cascades [93,94,95,96]. Secondary bile acids, such as deoxycholic acid and lithocholic acid, regulate the immune system by acting as ligands for TGR5, resulting in suppression of tumor necrosis factor-α production by macrophages [97]. Secondary bile acids have also been demonstrated to induce the expression of the transcription factor Foxp3, as well as the production of RORγ+ regulatory T-cells [98,99,100]. A recent study demonstrated a mechanism of food sensitization via bile acid-mediated activation of retinoic acid responsive element in dendritic cells, thereby promoting food-allergen specific IgE and IgG1 [101]. The role of bile acids in regulating immunological and inflammatory responses is complex, and continued studies will help elucidate their role in the development of therapeutic approaches to food allergies.

### 3.3. Sphingolipids

Emerging evidence supports the connection between sphingolipids, a class of lipids with a long-chain sphingoid base, to food allergy pathogenesis [102,103,104]. Sphingolipids may be ingested, produced endogenously, and, notably, synthesized by gut microbiota, such as Bacteroidetes [105]. In a metabolomic profiling study, Crestani et al. identified low levels of sphingolipids (such as sphingomyelin and ceramide) as a distinct hallmark for food allergy phenotypes [106]. Acid sphingomyelinase enzymatically converts sphingomyelin to ceramide, and its metabolic pathway has been linked to Th17 immune responses [107]. A recent study demonstrated that the administration of dietary glucosylceramide suppressed allergic responses in mice, and that the sphingoid base of this substance inhibited mast cell degranulation by binding a leukocyte mono-immunoglobulin-like receptor [108]. Although full details of the relationship between sphingolipid metabolism and gut dysbiosis-mediated food allergies remain unclear, this topic deserves further research and may aid the discovery of novel therapeutics.

### 3.4. Amino Acids

Although most amino acids are absorbed in the small intestine, some will remain in the alimentary tract and be available for metabolism by gut microbiota [109]. Metabolism of branched-chain amino acids results in the production of corresponding branched SCFAs, which, as with propionate and butyrate, also inhibit histone deacetylases to regulate gene expression [110]. Notably, dysregulation in the metabolism of lysine, leucine, and threonine have been associated with food allergies [106]. Emerging evidence suggests that there may be clinical benefits to combining symbiotics with amino acid formulas, and continued studies on amino acid supplementation for promoting beneficial microbial composition will be crucial [111,112,113].

The essential amino acid tryptophan is processed by gut lactobacilli into metabolites, such as indole-3-aldehyde, which activates the transcription factor ‘aryl hydrocarbon receptor’ in epithelial and immune cells that modulates Th17, as well as regulatory T-cell differentiation [114,115]. The aryl hydrocarbon receptor pathway has been shown to promote allergic inflammation in mouse models [116]. Moreover, a metabolomics study found that mice with ovalbumin-induced food allergies had increased tryptophan metabolism with higher levels of indole derivatives [117]. It has also been shown that *Bifidobacterium* supplementation results in decreased intestinal inflammation, and that it is negatively associated with levels of indole-3-lactic acid, another tryptophan metabolite [118]. A study aimed at identifying probiotic metabolites for application in allergic diseases specifically identified D-tryptophan as a promising therapeutic substance, as D-tryptophan supplementation in mice induced the production of lung and gut regulatory T-cells while attenuating T_H_2 responses [119]. These interactions between allergic responses and microbiota-driven tryptophan metabolism represent a unique opportunity for new therapies. For example, one study demonstrated the use of fructooligosaccharides to modulate gut microbiome composition and tryptophan metabolism to protect mice against ovalbumin-induced food allergies through the regulation of Th17/regulatory T-cell balance [120].

An improved understanding of the immunological consequences of these and yet uncharacterized metabolites will help support evidence-based dietary guidelines and inform novel metabolite-based therapeutic strategies.

## 4. Targeted Microbial Therapies

### 4.1. Fecal Microbiota Transplantation

Fecal microbiota transplantation (FMT) involves the transfer of microbial communities, for example from select healthy donors to recipients with gut dysbiosis. Indeed, FMT has been shown to be effective for treating *Clostridioides difficile* colitis and represents a promising novel therapeutic strategy for many disorders associated with perturbations of the gut microbiome [121,122,123]. Early experiments by Rodriguez et al. demonstrated that FMT from healthy human infants could protect against cow’s milk allergy in a gnotobiotic mouse model [124]. On the other hand, Rivas et al. later showed that increased susceptibility towards food allergies could be imparted onto germ-free mice via microbial transplantation [125]. These complementary results emphasize the critical role of the microbiome in food allergies and need for careful selection of FMT donors. Recent preclinical studies have continued to provide promising evidence that healthy human microbiota can protect against food allergy development in mouse models [30,31,126,127]. Feehley et al. demonstrated that transplantation of gut microbiota from healthy infants afforded protection against cow’s milk allergy to germ-free recipient mice sensitized to cow’s milk allergen [31]. Further studies by Abdel-Gadir et al. similarly showed that healthy donor FMT led to mitigation of food allergy response in mice with increased genetic susceptibility (Il4raF709), whereas this protective effect was not observed following FMT using infant donors with food allergies [30].

Clinical trials of FMT for food allergies are being conducted to build upon these promising results. One such study is a phase I open-label trial which recently completed enrollment as of September 2021. It aims to evaluate the safety and efficacy of oral encapsulated FMT for patients with peanut allergy (NCT02960074). Patients in this study will either undergo FMT alone or FMT with an antibiotic pre-treatment, and they will subsequently undergo a double-blind placebo-controlled food challenge with peanut protein.

Although FMT presents a rich avenue for investigating novel therapeutics against food allergies, the potential adverse effects of FMT must be rigorously considered. Severe side effects of FMT have been documented in the literature, such as drug-resistant microbial associated sepsis and unanticipated systemic immune responses [128,129]. Continued studies will, therefore, be imperative to better characterize the risks associated with FMT for food allergies, as well as to optimize donor and host characteristics for effective therapy.

### 4.2. Bacteriotherapy

A popular avenue for modulating the gut microbiome is through the introduction of specific bacterial strains thought to have direct beneficial properties and/or to promote healthy microbiome composition and function, in lieu of complete microbiome transplantation. Unlike probiotics, which are regulated as foods, these live microbial products are considered active pharmaceutical ingredients and would be subject to regulation by the FDA [130].

Determining which bacterial species would be good candidates for bacteriotherapy for food allergies requires determining the beneficial functionality of the species. Methods include assessing for differential abundance of bacterial species between affected and control groups, in combination with transcriptomics or metabolomics, to identify the microbiota with likely beneficial gene expression or metabolite production seen in healthy controls but not in affected subjects [131,132]. For example, Feehley et al. paired transcriptomics with differential microbiome composition in infants with and without cow’s milk allergy to identify a bacterial species that protected against food allergies [31]. They first determined which bacteria were enriched in healthy infants compared to those with cow’s milk allergy, then determined if the genes upregulated in the microbiota of mice colonized with healthy infant stool were also upregulated in mice colonized with the bacteria that was identified (*Anaerostipes caccae*) as enriched in healthy infants. Additional factors to consider in picking bacteriotherapy candidates include engraftment of the species in the gut microbiome, which may require antibiotic use prior to administration of bacteriotherapy, as well as prebiotics to encourage growth and retention of these species.

In the field of food allergies, various observational studies noted differences in gut microbiota in subjects who have developed food allergies compared to those who did not [30,32,34,36,133,134,135]. Not only were there compositional differences in the microbiota, but gene content and metabolomic signatures were also different, suggesting differential functionality of the microbiota between subjects with food allergies and without [31,32,106,133].

Various groups have tried to identify specific candidate microbial species with therapeutic potential against food allergies. Atarashi et al. found that a subset of regulatory T-cell-inducing *Clostridium* species (belonging to clusters IV, XIVa, and XVIII) conferred resistance to colitis and systemic immunoglobulin E response, and demonstrated a protective effect against ovalbumin-induced allergic diarrhea in mice following the oral administration of these *Clostridium* species [136,137]. Stefka et al. also showed that colonization of germ-free mice with *Clostridium* species from clusters IV and XIVa led to a protective effect against sensitization to peanut allergens [36]. Abdel-Gadir et al. found that multiple Clostridial taxa were affected in dysbiosis associated with food allergies, and they used bacteriotherapy with a consortium of six Clostridial species to successfully suppress food allergies in sensitized mice [30]. They also found they were able to suppress food allergies with phylogenetically distinct microbiota, specifically a consortium of five Bacteroidales species. Furthermore, they were also able to suppress food allergies using a single Clostridial bacterium (*Subdoligranulum variabile*). Similar to Abdel-Gadir’s group, Feehley et al. were able to identify, using transcriptomics, a single Clostridial species (*Anaerostipes caccae*) that afforded protection against cow’s milk allergy to germ-free recipient mice [31]. Bao et al. expanded on these findings, correlating differential bacterial abundance and metabolites in healthy as compared to allergic twins to identify an additional two Clostridial species (*Ruminococcus bromii* and *Phascolarctobacterium faecium*) that could be candidates for bacteriotherapy in food allergies [32].

### 4.3. Industry Developments and Ongoing Clinical Trials

A more limited number of microbiome-focused therapeutics are under investigation for food allergies than, for example, the treatment of *Clostridioides difficile* [130]. Current therapeutics with clinical trial registration (summarized in Table 1 and Table 2) include prebiotics, probiotics, synbiotics, bacteriotherapy, FMT, dietary interventions, and maternal microbiome transplant via vaginal seeding after a caesarean section.

One of the few clinical trials investigating bacteriotherapy for food allergies include a randomized double-blind phase I/II trial investigating VE416 (NCT03936998), an orally administered bacterial consortium created by Vedanta Bioscience, Inc., Cambridge, MA, USA, based on the candidate strains identified by Atarashi et al. [136,137]. Subjects with peanut allergy will be randomized into four groups encompassing combinations of VE416 with vancomycin pre-treatment along with corresponding placebos, and subsequently undergo double-blind placebo-controlled oral food challenge with peanut protein. Additionally, Siolta Therapeutics has a phase I/II trial in its ADORED study (NCT05003804) investigating a live biotherapeutic of intestinal bacteria from healthy donors (STMC-103H) to prevent the development of allergic disease in at-risk newborns.

There are various interventional trials utilizing prebiotics, probiotics, or synbiotics to investigate effects on food allergies. These include trials where prebiotics are used as adjuncts to oral immunotherapy, such as the Pinpoint Trial (NCT05138757) and the OPIA trial in Australia (ACTRN12617000914369), which utilizes a butyrylated high-amylase maize starch as the prebiotic. Various Australian trials include the addition of *Lactobacillus rhamnosus* probiotics to oral immunotherapy (ACTRN12616000322437, ACTRN12608000594325), and demonstrated improved sustained unresponsiveness in patients receiving probiotic and peanut oral immunotherapy compared to a placebo, but not compared to oral immunotherapy alone [138,139,140]. Probiotic *Lactobacillus rhamnosus* represents a rational therapeutic candidate, as it is historically known to be tolerated in early life and has been successful in preventing atopic diseases, such as eczema, asthma, and atopic rhinitis [141]. Completed trials from Canini et al. (ACTRN12610000566033; NCT01634490) evaluated the effect of extensively hydrolyzed casein formula supplemented with *Lactobacillus rhamnosus* GG on the development of tolerance in infants with cow’s milk allergy [68,142]. Combining prebiotics with probiotics (to make synbiotics) can be seen in the Synbiotic Cohort Study through Nutricia UK (NCT05046418), where synbiotics are added to hypoallergenic formula in infants with cow’s milk allergy. *Bifidobacterium* is another logical candidate as a potential therapeutic, as this genus is underrepresented in allergic infants during the early life period [143]. A completed trial from the Netherlands (NTR3979) found that a synbiotic-containing fructooligosaccharides and *Bifidobacterium breve* M-16V helped alter the gut microbial composition of non-IgE cow’s milk allergic infants to resemble more closely that of healthy infants [144,145]. Notably, another trial examining the same intervention (NTR3725) found no significant difference at 12 months or 24 months in cow’s milk tolerance following the addition of synbiotics [146].

**Table 1 nutrients-14-05155-t001:** Completed clinical trials of microbiome-related therapeutics for food allergies.

Therapeutic Strategy	Company or Organization	Clinical Trial Phase	Intervention Name	Intervention Description	Study Subjects	Primary Study Outcome	Findings	Trial Identifier	Trial Name
Biotherapeutics	Boston Children’s Hospital	Phase 1	N/A	Unspecified oral encapsulated microbiota transplantation	Adults 18–40 years with peanut allergy	Presence of fecal microbiota transplantation-related adverse events grade 2 or above	Not yet published	NCT02960074	Evaluating the Safety and Efficacy of Oral Encapsulated Fecal Microbiota Transplant in Peanut Allergic Patients
Pre/Pro/Synbiotic	Royal Children’s Hospital; Murdoch Children’s Research Institute; Prota Therapeutics	Phase 2b/3	PRT100	Probiotic *Lactobacillus rhamnosus* ATCC 53103; peanut oral immunotherapy	Children 1–10 years with peanut allergy	Sustained unresponsiveness to peanut protein by double-blind placebo-controlled food challenge	Sustained unresponsiveness at 12 months in 36/79 (46%) in the probiotic and peanut oral immunotherapy group vs. 42/85 (51%) in the peanut oral immunotherapy group vs. 2/39 (5%) in placebo group [138].	ACTRN12616000322437	A multicentre, randomised, controlled trial evaluating the effectiveness of probiotic and peanut oral immunotherapy (PPOIT) in inducing desensitisation or tolerance in children with peanut allergy compared with oral immunotherapy (OIT) alone and with placebo
	Royal Children’s Hospital	Phase 2b	NCC4007	Probiotic *Lactobacillus rhamnosus CGMCC 1.3724;* peanut oral immunotherapy	Children 1–10 years with peanut allergy	Sustained unresponsiveness to peanut protein by double-blind placebo-controlled food challenge	Sustained unresponsiveness after 2 to 5 weeks in 23/28 (82%) in the probiotic and peanut oral immunotherapy group vs. 1/28 (4%) in placebo [139]. Quality-of-life scores increased in the probiotic and peanut oral immunotherapy group (*n* = 19) but not the placebo group (*n* = 19), with a strong correlation between quality-of-life scores and frequency/amount of peanuts eaten up to the final endpoint at four years [140].	ACTRN12608000594325	Study of effectiveness of probiotics and peanut oral immunotherapy (OIT) in inducing desensitisation or tolerance in children with peanut allergy
	National Health and Medical Research Council; Sydney Children’s Hospital Network	N/A	Butyrylated high-amylase maize starch (HAMSB)	Prebiotic dietary fiber; peanut immunotherapy	Children 10–16 years with peanut allergy	Sustained unresponsiveness to peanut protein by double-blind placebo-controlled food challenge	Not yet published	ACTRN12617000914369	Oral peanut immunotherapy with a modified dietary starch adjuvant for treatment of peanut allergy in children aged 10–16 years
	Danone Nutricia Research	N/A	N/A	Amino acid formula with prebiotic oligofructose, prebiotic inulin, and probiotic *Bifidobacterium breve M-16V*	Infants <13 months with cow’s milk allergy	Cow’s milk tolerance by double-blind placebo-controlled food challenge	No significant difference in cow’s milk tolerance at 12 and 24 months [146].	NTR3725	A prospective double blind randomised controlled study to evaluate the immunological benefits and clinical effects of an elimination diet using an amino acid formula (AAF) with an added pre-probiotic blend in infants with Cow’s Milk Allergy (CMA)
	Danone Nutricia Research	N/A	N/A	Amino acid formula with prebiotic oligofructose, prebiotic inulin, and probiotic *Bifidobacterium breve M-16V*	Infants <13 months with suspected cow’s milk allergy	Fecal percentages of bifidobacteria and *Eubacterium rectale*/*Clostridium coccoides*	Experimental vs. placebo group had higher median percentage bifidobacteria and lower *Eubacterium rectale*/*Clostridium coccoides* at 8 weeks: (35.4% vs. 9.7%; *p* < 0.001), (9.5% vs. 24.2%; *p* < 0.001) respectively [144] and at 26 weeks: (47.0% vs. 11.8%; *p* < 0.001), (13.7% vs. 23.6%; *p* = 0.003) respectively [145].	NTR3979	An Amino Acid based Formula with synbiotics: Effects on gut microbiota diversity and clinical effectiveness in suspected gastrointestinal non-IgE mediated Cow’s Milk Allergy (ASSIGN I)
	University of Naples Federico II	N/A	Nutramigen LGG	*Lactobacillus GG* in extensively hydrolyzed casein formula	Children 1–24 months with cow’s milk allergy	Tolerance to oral food challenge	Group receiving *Lactobacillus GG* vs. control had more patients achieving tolerance to non-IgE mediated cow’s milk allergy at 6 months (16 vs. 6; *p* = 0.017), IgE mediated cow’s allergy at 12 months (5 vs. 1; *p* = 0.046), and non-IgE mediated cow’s milk allergy at 12 months (17 vs. 8; *p* = 0.006) [68].	ACTRN12610000566033	A randomised controlled trial on the effect of extensively hydrolyzed casein formula containing Lactobacillus GG (LGG) vs. extensively hydrolyzed casein formula on time of tolerance acquisition in children with cow’s milk allergy
	University of Naples Federico II	N/A	Nutramigen LGG	*Lactobacillus GG* in extensively hydrolyzed casein formula	Infants 1–12 months with cow’s milk allergy	Time to tolerance acquisition	Compared to groups receiving other formula types, there more patients who achieved tolerance to cow’s milk allergy in the groups receiving *Lactobacillus GG* with extensively hydrolyzed casein formula (78.9%; *p* < 0.05) and extensively hydrolyzed casein formula alone (43.6%; *p* < 0.05) [142].	NCT01634490	Effects of Different Dietary Regimens on Tolerance Acquisition in Children With Cow’s Milk Allergy

**Table 2 nutrients-14-05155-t002:** Ongoing clinical trials of microbiome-related therapeutics for food allergies.

Therapeutic Strategy	Company or Organization	Clinical Trial Phase	Intervention Name	Intervention Description	Study Subjects	Primary Study Outcome	Trial Identifier	Trial Name
Biotherapeutics	Massachusetts General Hospital; Vedanta Biosciences	Phase 1/2	VE416	Consortium of inactive commensals	Patients 12–55 years with peanut allergy	Number of patients with treatment related adverse events; peanut protein tolerance by double-blind placebo-controlled food challenge	NCT03936998	VE416 for Treatment of Food Allergy
	Siolta Therapeutics	Phase 1b/2	STMC-103H	Live biotherapeutic of unspecified intestinal bacteria	Children 1–6 years; 1–12 months; 0–7 days with immediate family history of allergic disorder	Frequency, type, and severity of adverse events; incidence of atopic dermatitis Secondary outcome: incidence of sensitization to food allergen; incidence of food allergies	NCT05003804	Allergic Disease Onset Prevention Study (adored)
	Evelo Biosciences	Phase 2	EDP1815	*Prevotella histicola*	Adults 18–75 years with atopic dermatitis	EDP1815 efficacy defined as >50% decrease in Eczema Area Severity Index	NCT05121480	A Study Investigating the Effect of EDP1815 in the Treatment of Mild, Moderate and Severe Atopic Dermatitis
Pre/Pro/Synbiotic	University of Chicago	Phase 1/2	N/A	Unspecified prebiotic	Children 4–7 years with peanut allergy	Peanut protein tolerance by double-blind placebo-controlled food challenge	NCT05138757	Pinpoint Trial: Prebiotics IN Peanut Oral ImmunoTherapy
	Chinese University of Hong Kong	N/A	N/A	Unspecified probiotic; peanut oral immunotherapy	Children 1–17 years with peanut allergy	Sustained unresponsiveness to peanut protein by double-blind placebo-controlled food challenge	NCT05165329	A Randomised, Controlled Trial Evaluating the Effectiveness of Probiotic and Peanut Oral Immunotherapy (PPOIT) in Inducing Desensitisation or Remission in Chinese Children With Peanut Allergy Compared With Oral Immunotherapy (OIT) Alone and With Placebo
	Massachusetts General Hospital; Mead Johnson Nutrition	Phase 2 (Terminated)	N/A	Extensively hydrolyzed casein formula; *Lactobacillus GG*; Amino acid formula	Infants up to 120 days with suspected cow’s milk allergy	Tolerance to cow’s milk protein	NCT02719405	Impact of Infant Formula on Resolution of Cow’s Milk Allergy
	Johnson & Johnson; Evolve BioSystems	N/A	Evivo	*Bifidobacterium longum* subspecies *infantis* strain EVC001	Healthy infants up to 14 days	Number of subjects with atopic dermatitis Secondary outcome: Percentage of subjects with *Bifidobacterium Infantis* gut colonization	NCT04662619	A Study of a Probiotic Food Supplement Containing B. Infantis (EVC001) in Healthy Breastfed Infants at Risk of Developing Atopic Dermatitis
	NovoNatum	N/A	BioAmicus Lactobacillus drops	*Lactobacillus Reuteri* NCIMB 30351	Children 1–5 months with colic, constipation, diarrhea, or regurgitation	Change in number with infantile colic Secondary outcome: presence of skin or food allergies; stool 16 s RNA sequencing	NCT04262648	Randomized Placebo-controlled Study of L. Reuteri NCIMB 30351 in GI Functional Disorders and Food Allergy in Newborns
	Nutricia UK	N/A	N/A	Hypoallergenic formula containing unspecified synbiotics	Infants <13 months with confirmed or suspected cow’s milk allergy	Healthcare utilization by electronic health records Secondary outcome: Clinical outcomes related to cow’s milk allergy	NCT05046418	Synbiotics Cohort Study
Vaginal Seeding	National Institute of Allergy and Infectious Diseases; Immune Tolerance Network; Pharmaceutical Product Development; Rho Federal Systems Division	Phase 1	N/A	Maternal vaginal microbiota	Neonate of adult female 18–45 with first degree relative with atopic disease or food allergies	Presence of sensitization to at least one food allergen	NCT03567707	Vaginal Microbiome Exposure and Immune Responses in C-section Infants
	Karolinska Institutet; Uppsala University Linkoeping University Umeå University Örebro University	N/A	N/A	Maternal vaginal and fecal microbiota	Neonate of healthy adult mother	Incidence of immunoglobulin E-associated allergic disease	NCT03928431	Restoration of Microbiota in Neonates (RoMaNs)

Some microbiome therapeutics which may affect food allergy development via a more indirect route include therapeutics to prevent or treat atopic dermatitis, which is a risk factor for the development of food allergies and tends to precede food allergy onset [14,15,17,147]. Ongoing trials of bacteriotherapy and probiotics include Evelo Biosciences’ EDP1815, a single strain of *Prevotella histicola* (NCT05121480), and Evolve BioSystems’ Evivo probiotic containing the *Bifidobacterium longum* subspecies *infantis* strain EVC001 (NCT04662619). *Prevotella* species represent a particularly promising target. as recent evidence demonstrated that maternal carriage of *Prevotella* is associated with a strong protection against the development of food allergies [148].

Additionally, transplanting maternal vaginal microbiota onto neonates delivered via C-section is thought to affect the trajectory of skin and gut bacterial colonization in infants, as differential gut microbial composition has been observed in infants delivered vaginally vs. via C-section [149,150]. A few vaginal seeding trials were identified (NCT03567707, NCT03928431).

Microbiome-based therapeutics that may soon proceed to clinical trials include ClostraBio’s microbiome-modulating bioactive polymer combined with a butyrate-producing live biotherapeutic (*Anaerostipes caccae*) [31,132].

## 5. Conclusions

Exciting data from human association studies and preclinical models have provided compelling evidence for a role for gut microbiota and their metabolites in food allergy pathogenesis. These insights have motivated a wide range of recently completed and ongoing clinical trials targeting the microbiome including dietary interventions, prebiotics, probiotics, SCFAs, FMT, and bacteriotherapy. Given pre-clinical findings that showed that FMT and bacteriotherapy appear to suppress food allergies in mice, it will be exciting to see the results of these studies in humans. If successful, these microbiome therapeutics may become the first truly disease-modifying treatments for food allergies. Results from these clinical studies will provide insight into what challenges lie ahead and a foundation for further clinical trials to elucidate which methods will be most efficacious for food allergies.

## Figures and Tables

**Figure 1 nutrients-14-05155-f001:**
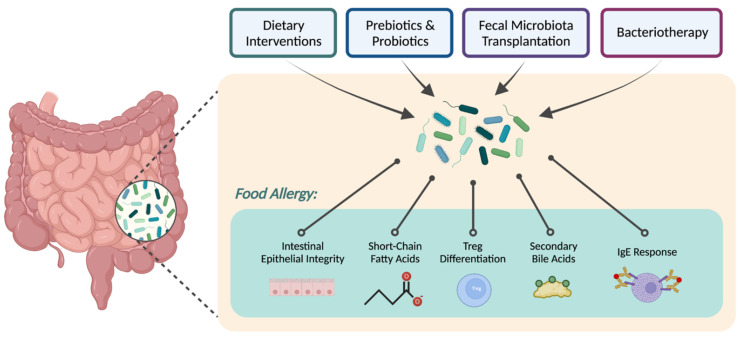
Outline of microbiome-related therapeutic strategies for food allergies.

## Data Availability

Not applicable.

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
