# Peer review of "Microbiome Therapeutics for Food Allergy"

_nutrients, 2022, doi:10.3390/nu14235155_

Round 1

Reviewer 1 Report

The author systemically reviewed microbiome-related approaches to prevent or treat food allergy and give insight in a future clinical practice as gut microbiome strategy. The manuscript was well addressed, major points are as below: 

Major points:

1.      I would like to ask the author to reorganize abstract for better interpreting this review.

2. In the introduction, the author may add more information on the interaction between gut microbiota and food allergy.

3. I would like to suggested more figures that help to better illustrate the mechanisms of gut microbial function in food allergy.

4. The content of gut microbiota is not sufficient.

5. The conclusion should list more perspective in this field.

Reviewer 2 Report

The authors attempted to summarize past research studies developed to focus on the role of human gut microbiomes in preventing and treating food allergies. They pointed out that the increasing prevalence of food allergy is believed to be a combination of genetic and environmental factors. Among them, gut microbial compositions have been shown to be an important mediator: gut microbiomes are influenced by dietary patterns, while their compositions strongly modulate immune tolerance. Therefore, microbiome-based therapeutics such as dietary intervention or probiotics introduction can be leveraged to prevent or treat food allergies and associated diseases. They summarized three potential mechanisms through which gut microbiomes can be used as a modulator or cure: (1) dietary intervention, (2) promoting health-benefiting metabolites and modulating immunomodulatory metabolites gut microbes produce, and (3) microbial therapies including bacteriotherapy or probiotics. I am quite open to looking at a revised version if the authors could address some major and minor issues in a satisfactory fashion, which we describe in more detail below.

Major issues:

1.     My major concern is that authors only focus on microbiome-related treatments without mentioning other treatment methods such as drug treatments developed for food allergies. It would be great if authors can compare the efficiency of microbiome-related treatments with other treatments and discuss the benefits and disadvantages.

2.     It is great that authors can summarize completed and ongoing clinical trials of microbiome-related therapeutics and pointed out the trial identifiers. I would appreciate more if authors can provide citations to published papers or website information so that readers can easily find those studies.

3.     The authors mentioned branched-chain SCFAs in Section 3.4. Are they referring to “branched short-chain fatty acids (BSCFAs)”?

Minor comments:

1.     The citations should appear before the commas or periods. For example, authors wrote “Food allergies have been rising in prevalence in recent decades and are also the most common cause of anaphylaxis in children.[1, 2]”. It should be “Food allergies have been rising in prevalence in recent decades and are also the most common cause of anaphylaxis in children [1, 2].”

2.     “GPR41-dependent manor.” -> “GPR41-dependent manner.”

3.     “…and subsequently undergo double-blind placebo controlled oral food challenge” -> “…and subsequently undergo a double-blind placebo-controlled oral food challenge”

Round 2

Reviewer 2 Report

The authors answer all my concerns very well. I have no other comments.